Isolated teeth of Anhangueria  (Pterosauria: Pterodactyloidea) from the Lower Cretaceous of Lightning Ridge, New South Wales, Australia

Brougham Tom tbrougha@myune.edu.au 1
Smith Elizabeth T. 2
Bell Phil R. 1
1 School of Environmental and Rural Science, University of New England , Armidale , New South Wales , Australia
2 Australian Opal Centre , Lightning Ridge , New South Wales , Australia
Farke Andrew
Electronic publication date: 2017 May 3
Publication date: 2017
Volume: 5
Electronic Location ID: e3256
Received 2017 Jan 11; Accepted 2017 Mar 31
Copyright: ©2017 Brougham et al.
Copyright year: 2017
Copyright holder: Brougham et al.
License: This is an open access article distributed under the terms of the Creative Commons Attribution License, which permits unrestricted use, distribution, reproduction and adaptation in any medium and for any purpose provided that it is properly attributed. For attribution, the original author(s), title, publication source (PeerJ) and either DOI or URL of the article must be cited.
License URL: https://creativecommons.org/licenses/by/4.0/

Keywords: Pterosauria, Cretaceous, Australia, Teeth

Funding: Australian Postgraduate Award Australian Research Council Discovery Early Career Researcher Award DE170101325 TB was funded by an Australian Postgraduate Award. PRB was funded under an Australian Research Council Discovery Early Career Researcher Award (project ID: DE170101325). The funders had no role in study design, data collection and analysis, decision to publish, or preparation of the manuscript.

==============================
The fossil record of Australian pterosaurs is sparse, consisting of only a small number of isolated and fragmentary remains from the Cretaceous of Queensland, Western Australia and Victoria. Here, we describe two isolated pterosaur teeth from the Lower Cretaceous (middle Albian) Griman Creek Formation at Lightning Ridge (New South Wales) and identify them as indeterminate members of the pterodactyloid clade Anhangueria. This represents the first formal description of pterosaur material from New South Wales. The presence of one or more anhanguerian pterosaurs at Lightning Ridge correlates with the presence of ‘ornithocheirid’ and Anhanguera-like pterosaurs from the contemporaneous Toolebuc Formation of central Queensland and the global distribution attained by ornithocheiroids during the Early Cretaceous. The morphology of the teeth and their presence in the estuarine- and lacustrine-influenced Griman Creek Formation is likely indicative of similar life habits of the tooth bearer to other members of Anhangueria.

Introduction

Pterosaurs first appeared in the Late Triassic and diversified rapidly into the Jurassic. At the peak of their diversity in the Cretaceous, pterosaurs where present on all continents, including Antarctica (Barrett et al., 2008; Upchurch et al., 2015). During the Early Cretaceous, ornithocheiroid pterosaurs in particular achieved an essentially global distribution and are known from remarkably complete specimens discovered from Lagerstätten in South America and China (Upchurch et al., 2015).

Figure 1 Australian Cretaceous pterosaur occurrences.

The extent of Cretaceous Eromanga and Surat basins in the early to middle Albian is represented by the grey area separated by dashed line, and the epicontinental Eromanga Sea is represented by the area in blue. Locations of pterosaur occurrences (marked by circles) represent: (1) Giralia Range (Miria Formation, Maastrichtian); (2) Gingin (Molecap Greensand; Cenomanian–Turonian); (3) Hughenden (Mackunda and Toolebuc formations; Albian); (4) Boulia (Toolebuc Formation; Albian); (5) Dinosaur Cove (Otway Group; Aptian–Albian); and (6) Lightning Ridge (Griman Creek Formation, Albian). The inset map shows the area in the vicinity of Lightning Ridge (location 6) and the locations of the two new Australian pterosaur occurrences (marked by triangles). Australia coastline from GEODATA COAST 100K 2004 (http://www.ga.gov.au/metadata-gateway/metadata/record/61395); basin extents data from Australian Geological Provinces, 2013.01 edition (http://www.ga.gov.au/metadata-gateway/metadata/record/74371/); both released by Geoscience Australia under CC BY 4.0 licence (https://creativecommons.org/licenses/by/4.0/). Eromanga Sea extent uses data taken from the palaeoshoreline shapefiles of Heine, Yeo & Müller (2015) (https://github.com/chhei/Heine_AJES_15_GlobalPaleoshorelines), released under CC BY 4.0 licence (https://creativecommons.org/licenses/by/4.0/).

By contrast, the fossil record of pterosaurs in Australia is very sparse and composed solely of isolated and fragmentary remains from the Cretaceous of Queensland, Victoria and Western Australia (Fig. 1). The taxonomic status of Australia’s record of Cretaceous pterosaurs has been reviewed recently and comprehensively by Fletcher & Salisbury (2010), and also by Kellner, Rodrigues & Costa (2011). Following the pterosaur phylogeny of Andres, Clark & Xu (2014), material representative of three clades of pterodactyloid pterosaurs has been identified from Australia: pteranodontoids (Molnar & Thulborn, 1980; Molnar & Thulborn, 2007; Molnar, 1987; Kellner et al., 2010; Kellner, Rodrigues & Costa, 2011); ctenochasmatoids (Fletcher & Salisbury, 2010); and azhdarchids (Bennett & Long, 1991; see Fig. 2). The pteranodontoid-dominated horizons of the Albian Toolebuc Formation near Boulia and Hughenden in central-western Queensland have been the most productive sites for Australian pterosaurs to date (Fig. 1). The only known Australian ctenochasmatoid was found in the slightly younger Mackunda Formation near Hughenden. Late Cretaceous pterosaur occurrences are restricted to the Perth and Carnarvon basins of Western Australia, the latter of which is the source of the only known azhdarchid remains from Australia. A purported pterosaur tibiotarsus from the Lower Cretaceous Otway Group of southern Victoria (Rich & Rich, 1989), and reinterpreted by Bennett & Long (1991) as a metatarsus, has been mentioned but not described.

Figure 2 Chrono- and lithostratigraphic context of Australian pterosaur occurrences.

Formally documented pterosaur occurrences within Australia are restricted to the Albian of Queensland and New South Wales (anhanguerians and ctenochasmatids) and the Cenomanian-Turonian and Maastrichtian of Western Australia (anhanguerians and azhdarchids). Australian basin lithostratigraphic data from the Geoscience Australia Datapack for TimeScale Creator (http://data.gov.au/dataset/dec45071-11a4-4d28-92a6- 5d8dc9e5d978). Silhouettes provided courtesy of Phylopic (http://phylopic.org); Azhdarchidae by Darren Naish (vectorised by T Michael Keesey), Ctenochasmatidae courtesy of Jaime Headden, both released under CC BY 3.0 licence (http://creativecommons.org/licenses/by/3.0/); Silhouette for Anhangueria modified from Claessens, O’Connor & Unwin (2009, fig. 3d).

Pterosaur teeth in Australia are known only from those that remained within the jaw of the probable pteranodontoid Mythunga camara (Molnar & Thulborn, 2007, fig. 2), and from an isolated tooth associated with an ‘ornithocheirid’ mandible (Fletcher & Salisbury, 2010, figs. 3I–3J). No pterosaur material from New South Wales has to date been described. Smith (1999) figured two purported pterosaur long bones from the Lower Cretaceous Griman Creek Formation at Lightning Ridge, but was provided without a systematic description. Here, we describe two isolated pterosaur teeth from the same location, which constitute the first formal identification of material belonging to this clade of reptiles from New South Wales.

Locality and Geological Setting

The teeth were excavated from underground opal mines in the vicinity of Lightning Ridge, central-northern New South Wales, Australia (Fig. 1). Fossil- and opal-bearing rocks in the Lightning Ridge area are confined to the Lower Cretaceous Griman Creek Formation, situated in the Surat Basin that extends over parts of south-eastern Queensland and northern New South Wales. Together with the neighbouring Eromanga Basin, these form the majority of the present day Great Artesian Basin (GAB). The Griman Creek Formation is composed of thinly laminated and interbedded fine- to medium-grained sandstones, siltstones and mudstones, with carbonate cements, intraformational conglomerate beds and coal deposits (Burger, 1980; Green et al., 1997). Within the Griman Creek Formation, opal and fossils occur within interbedded siltstone and mudstone layers, often referred to as the Finch clay facies (Byrnes, 1977). Palynological evidence indicates that the Griman Creek Formation is associated with the Coptospora paradoxa Zone and correlates to the middle Albian (Burger, 1980). Apatite fission-track analyses on grains derived from core samples of the Queensland extent of the Griman Creek Formation indicate an upper age boundary of approximately 107 Mya (Raza, Hill & Korsch, 2009).

The depositional environment of the Griman Creek Formation is interpreted as a lacustrine to estuarine coastal floodplain with fluvial and deltaic influences (Bell et al., 2015). The area in the vicinity of Lightning Ridge was located at the south-eastern edge of the epicontinental Eromanga Sea that extended over much of central Australia during the Aptian and Albian (Frakes et al., 1987; Dettmann et al., 1992; Fig. 1). The Eromanga Sea was poorly connected to the open ocean as indicated by an invertebrate fauna composed almost entirely of species adapted to fresh water (Byrnes, 1977; Hocknull, 2000), coquina beds in the lower section of the Griman Creek Formation dominated by brackish and freshwater taxa (Green et al., 1997) and the lack of carbonate sediments (Rey, 2013). Cessation of sedimentation in and the onset of uplifting of the Surat and Eromanga Basins in the late Early Cretaceous is currently hypothesised to have led to the formation of opal beds in many areas of the GAB through erosion and oxidation of volcaniclastic sediments deposited between 130-95 Mya into in a cold, oxygen-deprived fluvial-deltaic environment (Rey, 2013).

The Griman Creek Formation at Lightning Ridge arguably contains the most abundant fossil record of Cretaceous terrestrial fauna in Australia (Dettmann et al., 1992), with crocodylomorphs (Etheridge, 1917; Molnar, 1980; Molnar & Willis, 2000), australosphenidian mammals (Archer et al., 1985; Rich, Flannery & Archer, 1989; Flannery et al., 1995), ornithopod dinosaurs (Molnar & Galton, 1986), megaraptoran theropods (White et al., 2013; Bell et al., 2015), enantiornithine birds (Molnar, 1999), plesiosaurs (Kear, 2006a), turtles (Smith, 2010; Smith & Kear, 2013), dipnoan lungfish (Kemp & Molnar, 1981; Kemp, 1993; Kemp, 1997) and a possible synapsid (Clemens, Wilson & Molnar, 2003) in addition to numerous species of non-marine macro-invertebrates (Byrnes, 1977; Hocknull, 2000; Kear & Godthelp, 2008; Hamilton-Bruce & Kear, 2010) and plants. Preservation of fossils at Lightning Ridge—including those specimens described here—is commonly in the form of natural casts, or pseudomorphs, in non-precious opal (e.g., Molnar & Willis, 2000; Clemens, Wilson & Molnar, 2003; Bell et al., 2015). The opalisation of both vertebrate and invertebrate fossils appears to have been a secondary process that occurred after initial permineralisation (Pewkliang, Pring & Brugger, 2008; Rey, 2013); however, fine-scale microstructural features of vertebrate bone such as trabeculae are sometimes observed in opalised specimens (TB, ES & PB, pers. obs., 2016).

Systematic Palaeontology

The following descriptions and discussion of pterosaur taxa follow the comprehensive pterosaur phylogeny of Andres, Clark & Xu (2014). This analysis differs most noticeably from another recent pterosaur phylogeny (Lü et al., 2012) in the presence of a monophyletic Archaeopterodactyloidea (sensu Kellner, 2003) and the inclusion of Ornithocheirus, Pteranodon and other closely related taxa within Lophocratia (sensu Unwin, 2003). Anatomical terminology for orientation of teeth follows that of Smith & Dodson (2003). Terminology for crown morphometrics follows that of Smith, Vann & Dodson (2005), whereas terminology for tooth enamel ornamentation follows that outlined by Hendrickx, Mateus & Araújo (2015) for theropods.

Pterosauria Kaup 1834	
Pterodactyloidea Plieninger 1901	
Ornithocheiroidea Seeley 1870	
Anhangueria Rodrigues & Kellner, 2013	

Material

The teeth (LRF 759 and LRF 3142) are preserved as isolated crowns, missing the roots and with eroded distal tips.

Locality

LRF 759 was excavated in the 1970s from an underground mineral claim at ‘Holden’s Four Mile’ opal field, approximately 4 km south west of Lightning Ridge (Fig. 1). LRF 3142 was excavated in 2015 from an underground mineral claim at ‘Dead Cat’ opal field, an extension of ‘Grannys Flat’ on the Coocoran opal fields, approximately 24 km west of Lightning Ridge (Fig. 1).

Preservation

Both LRF 759 and LRF 3142 are isolated tooth crowns with eroded apices; LRF 759 is also missing a portion of the distal part of the crown near the base. Both teeth are preserved as translucent potch, a form of non-precious opal; in LRF 759 the potch displays mauve play of opal colour, whereas LRF 3142 contains areas of dark grey within honey-coloured potch. In LRF 759, the translucency of the potch reveals a thin-walled basal cavity that has been infilled with a body of purple opal and buff-coloured mudstone (Fig. 3); the same area of LRF 3142 is infilled with white mudstone. These infills likely represent the extent of the tooth’s pulp cavity in each specimen. The preserved apex of LRF 3142 is gently rounded and forms a ‘cap’ that is delineated from the rest of the crown by a groove on the lingual surface (Fig. 4C) and by a ridge on the labial surface (Fig. 4E). This is unlikely to reflect the morphology of the original tooth considering the otherwise gentle tapering of the crown in both mesial-distal and labial-lingual planes. Taphonomic erosion and distortion of the apex through breakage or fracture prior to opalisation may be the cause of this feature, and its presence does not impact upon the preferred taxonomic placement of LRF 3142.

Figure 3 Tooth of Anhangueria indet. LRF 759.

(A, C) lateral, (B) mesial, (D) distal and (E) basal views. Scale bar equals 10 mm. Photo credit: Phil Bell.

Figure 4 Tooth of Anhangueria indet. LRF 3142.

(A) lingual view in reflected light and (B) transmitted light, (C) mesial, (D) labial, (E) distal and (F) basal views. Scale bar equals 10 mm. Photo credits: (A, C–F) by Tom Brougham; (B) by Robert A. Smith.

Description

LRF 759 (Fig. 3) has an elongate crown and oval basal cross-section as described below for LRF 3142 (Table 1). The lateral surfaces are evenly convex; it is not possible to distinguish labial and lingual surfaces. The crown also has a slight distal recurvature although it is less marked in comparison to that of LRF 3142; the distal margin is almost straight in lateral view and there is no lateral deflection of the crown towards the apex. There are no carinae on either the mesial or distal surfaces of the crown. The distal surface is flatter than the mesial surface.

Table 1 Dimensions of the Lightning Ridge pterosaur teeth.

Sample	Crown height  (mm)	Crown base length  (mm)	Crown base width  (mm)	
LRF 759	18.2	5.9	4.3	
LRF 3142	20.6	7.0	5.2	

Unlike LRF 3142, in LRF 759 the tooth crown is ornamented by longitudinal grooves extending essentially apicobasally along the surface (Fig. 5). A series of pits and shorter longitudinal grooves form a transverse band near the preserved base of the crown on one side (Fig. 5A), while weak and discontinuous striae narrower and shorter than the longitudinal grooves are present towards the apex. On the other surface (Fig. 5B), a faint longitudinal groove extends along almost the entire length of the preserved crown, with additional grooves constrained to the apical portion of the crown and approaching the mesial surface. The grooves and ridges all become more pronounced towards the apical end of the crown. On the same side, two deeply incised grooves extend almost parallel to each other from the preserved base of the crown, becoming deeper apically and converging at approximately one third of the way from the preserved apical end.

Figure 5 A cast of LRF 759 coated with ammonium chloride in lateral views.

lg, longitudinal groove; p, pits; s, striae. Scale bar equals 10 mm. Photo credit: Tom Brougham.

LRF 3142 (Fig. 4) is a gently recurved and elongated crown with a preserved height at least four times that of the width at the base. It is slightly longer mesiodistally than labiolingually wide at the base (Table 1; Fig. 4F) . The crown is slightly deflected apically such that one lateral surface is slightly convex in mesial view while the other is slightly concave (Fig. 4C). These surfaces are interpreted to be labial and lingual respectively following previous reports of isolated anhanguerian teeth (e.g., Wellnhofer & Buffetaut, 1999). The labial and lingual surfaces are convex, the labial slightly more so than the lingual, and meet mesially and distally to form carinae. The mesial carina is more clearly defined than the distal carina, and is slightly displaced lingually. The distal carina transitions from an acute point on the apical half of the crown to a gently curved edge on the more basal portion of the crown. No denticles are present on either the mesial or distal carinae. The tooth crown is smooth and ornamented by very fine irregularly-spaced apicobasal striae that are more clearly visible in transmitted light (Fig. 4B).

Discussion

Taxonomic identification

Elongate, conical teeth similar in morphology to those described above have been previously reported from Lightning Ridge, and include plesiosaurs (Kear, 2006b), ichthyosaurs (Kear, Boles & Smith, 2003), theropods (Bell et al., 2015) and crocodylians (Molnar, 1980). Other contemporaneous vertebrates that have been reported elsewhere from Australia that also bear similar teeth include pterosaurs (Molnar & Thulborn, 2007), teleost fish (Lees & Bartholomai, 1987; Berrell et al., 2014) and ichthyosaurs (Kear, Boles & Smith, 2003). The dental morphology of these groups is reviewed in brief below and compared with LRF 759 and 3142 to establish the basis for their assignment to Pterosauria.

Exclusion from Teleostei

The ichthyodectiform actinopterygians Cooyoo australis (Lees & Bartholomai, 1987) and Cladocyclus geddes (Berrell et al., 2014), both from the Albian of central Queensland have simple, conical and elongate teeth averaging only a few millimetres in height, with the dentary teeth of Cladocyclus also displaying a slight distal recurvature. The teeth are unornamented and do not bear any carinae on either the mesial or distal surfaces of the crown, unlike the condition in LRF 3142. The teeth of saurodontids have short, labiolingually-compressed triangular crowns and serrated carinae and have previously been mistaken for those of pterosaurs, particularly istiodactylids (e.g., Mkhitaryan & Averianov, 2011; Vullo, Buffetaut & Everhart, 2012). The Lightning Ridge teeth contrast strongly with those of saurodontids in their tall, elongate and slightly distally recurved crowns.

Exclusion from Plesiosauria

Plesiosaurs were ubiquitous in marine and marginal marine environments in Australia during the Lower Cretaceous (Kear, 2005a; Kear, 2005c; Kear, 2006b; Kear, 2006a). LRF 759 and LRF 3142 differ from previously described Australian plesiosaur teeth in the overall morphology of the tooth crown, development of carinae and enamel ornamentation. Plesiosaur teeth are typically elongate, lingually curved cones with a circular to ovoid basal cross-section. The mesial and distal surfaces of the crown lack carinae and have an apicobasally fluted enamel texture restricted to the lingual side of the crown, with flutes often bifurcating towards the base (Kear, 2005a, fig. 3f and 4b, Kear, 2005c; Kear, 2006b, figs. 2a–2g), although isolated teeth of Opallionectes lack any form of surface ornamentation (Kear, 2006a, text-fig. 2a).

Exclusion from Ichthyosauria

Only one valid species of ichthyosaur from Australia is presently recognised: Platypterygius longmani from the Albian Toolebuc Formation of central Queensland (Wade, 1990). P. longmani is known from an exceptionally preserved and articulated skull, complete with dentition. The teeth of P. longmani, and ichthyosaurs in general, differ from LRF 759 and LRF 3142 in the more robust and distally unrecurved crown with little or no labiolingual compression, a subcircular basal cross section and the presence of a fluted enamel texture that extends from near the tip of the crown down towards the base (Kear, 2005b, fig. 16).

Exclusion from Theropoda

The majority of unambiguous theropod remains from Australia have been referred to the recently diagnosed clade Megaraptora (Benson, Carrano & Brusatte, 2010). The dentition of megaraptorans is known in Australia from in situ and isolated teeth of the early Late Cretaceous Queensland theropod Australovenator wintonensis (Hocknull et al., 2009; White et al., 2015), as well as isolated teeth from the Aptian–Albian of the south coast of Victoria (Benson et al., 2012) and undescribed teeth from the Albian of Lightning Ridge (Smith, 1999; TB, ES & PB, pers. obs., 2016). These teeth are of the ziphodont type (sensu Hendrickx, Mateus & Araújo, 2015), that is strongly labiolingually-compressed, distally recurved and bearing denticulate distal carinae. Megaraptoran dentition is further characterised by pronounced labial and lingual depressions on the roots that extend onto the basal portion of the crown, such that the cross-section of the base of the crown has a ‘figure-eight’ shape (Novas, Ezcurra & Lecuona, 2008; Porfiri et al., 2014; White et al., 2015; Coria & Currie, 2016).

Teeth described as ‘conidont’ (sensu Hendrickx, Mateus & Araújo, 2015), similar to LRF 759 and LRF 3142, are present within theropods, most notably in spinosaurids. Spinosaurids are purportedly represented in Australia by a cervical vertebra from the Aptian–Albian Eumerella Formation (Barrett et al., 2011) but teeth are as yet unknown. The basal cross-section of baryonychine teeth is subcircular (e.g., Baryonyx; Charig & Milner, 1997) and differs from the oval basal cross-section typical of spinosaurine teeth (Richter, Mudroch & Buckley, 2012); however, in Spinosaurus, the shapes of the dentary alveoli transition from circular at the anterior end to more mesiodistally elongate and ovoid posteriorly (Stromer, 1915). Spinosaurid crowns often display a slight lingual curvature of the crown (Kellner & Mader, 1997; Richter, Mudroch & Buckley, 2012). Mesial and distal carinae in baryonychine teeth are ornamented by very fine serrations (e.g., Baryonyx, Suchomimus; Charig & Milner, 1997; Sereno et al., 1998) whereas the carinae of spinosaurines lack serrations entirely (e.g., Spinosaurus, Irritator; Stromer, 1915; Sues et al., 2002). The enamel of the crown in spinosaurid teeth appears granular and finely wrinkled with apicobasal fluting (see Hendrickx, Mateus & Araújo, 2015, fig. 4H, 6C–6D) that is usually more deeply impressed in baryonychines compared to spinosaurines (Stromer, 1915; Charig & Milner, 1997). However, baryonychine teeth have been reported with smooth enamel that is devoid of apicobasal flutes (Hone, Xu & Wang, 2010).

LRF 759 and LRF 3142 are distinct from megaraptorid teeth, and from ziphodont theropod teeth in general, in the oval basal cross-section of the crown, the slight degree of labiolingual compression of the crown, the apicobasal elongation of the crown, the lack of denticulated carinae and the absence of lingual or labial depressions at the base of the crown. LRF 759 and LRF 3142 are similar to spinosaurid teeth in their conical, elongate and slightly distally recurved crowns, and in the case of LRF 3142, the slight lingual curvature of the crown. However, they differ from the teeth of spinosaurines and baryonychines in lacking distinct fluting on either the labial or lingual surfaces. LRF 3142 has only very weak enamel ornamentation, but it is not certain if this is representative of the original enamel surface or a taphonomic artefact. The fine, discontinuous and irregularly-spaced longitudinal grooves of LRF 759 are unlike the enamel ornamentation of any known spinosaurid. In summary, the combination of features presented above for LRF 759 and LRF 3142 are inconsistent with spinosaurid dentition, and theropod dentition more broadly.

Exclusion from Crocodyliformes

Cretaceous crocodyliforms in Australia are rare and known only from an almost complete and articulated specimen of the neosuchian Isisfordia duncani from the upper Albian of central Queensland (Salisbury et al., 2006) and isolated skeletal material, including teeth, from Lightning Ridge (Etheridge, 1917; Molnar, 1980; Molnar & Willis, 2000). The teeth of Isisfordia are labiolingually compressed and distally unrecurved with distinct flutes extending along the crown (Salisbury et al., 2006, fig. 4F), whereas those from Lightning Ridge are conical and distally unrecurved with weak carinae (Molnar, 1980; Molnar & Willis, 2000).

The morphology of crocodyliform teeth, particularly those from the Mesozoic, displays considerable variation in terms of the degree of apicobasal elongation, mesiodistal curvature, acuteness of the apex, labiolingual compression, basal cross-sectional shape, presence and mode of development of carinae and denticles, and the presence and form of enamel ornamentation (Prasad & De Broin, 2002). In addition, many crocodyliform taxa display variation in tooth morphology along the premaxillary-maxillary and dentary tooth rows, while others retain a homodont dentition with variation, if any, only in the relative size of the tooth crowns. A homodont dentition of simple conical teeth appears in protosuchids, tethysuchians, paralligatorids, atoposaurids, and teleosaurs (e.g., Michard et al., 1990; Pol & Norell, 2004; Jouve, 2005; Young et al., 2014b; Tennant, Mannion & Upchurch, 2016). Thalattosuchian and some goniopholid teeth display a slight distal recurvature of the crown (e.g., Eutretauranosuchus, Machimosuchus; Smith et al., 2010; Young et al., 2014b). The remaining crocodyliform groups are heterodont to some degree. This may take the form of simple anterior-posterior morphological differentiation (e.g., Wannchampsus; Adams, 2014). More complex heterodonty occurs in notosuchians such as Notosuchus and Araripesuchus and the neosuchian Theriosuchus, in which at least three distinct tooth morphologies are present (Lecuona & Pol, 2008; Sereno & Larsson, 2009; Young et al., 2016).

Carinae are widely present on the dentition of crocodyliforms, with only a few exceptions (e.g., Eutretauranosuchus, Smith et al., 2010). Serrated carinae characterises the notosuchians, peirosaurids, Theriosuchus, paralligatorids, basal tethysuchians and thalattosuchians (e.g., Gasparini, Chiappe & Fernandez, 1991; De Lapparent De Broin, 2002; Schwarz & Salisbury, 2005; Sereno & Larsson, 2009; Andrade et al., 2010; Adams, 2014). Enamel ornamentation in crocodyliforms is typically in the form of flutes, and is present most notably in notosuchians, paralligatorids, goniopholids, Theriosuchus, basal eusuchians, tethysuchians and teleosaurids (e.g., Salisbury et al., 1999; Jouve, 2005; Schwarz & Salisbury, 2005; Delfino et al., 2008; Sereno & Larsson, 2009; Adams, 2014; Young et al., 2014b). In addition to or in place of flutes, fine anastomosing enamel textures are present in some tethysuchians, goniopholids and teleosaurs (e.g., De Lapparent De Broin, 2002; Andrade et al., 2011; Young et al., 2014a).

Some characteristics of crocodyliform teeth as reviewed above can be observed in LRF 759 and LRF 3142, such as the presence of unserrated carinae and slight labiolingual compression and distal recurvature of the crown. However, the confluence of the above characters is rarely present in any one crocodyliform taxon, and the comparatively smooth surface of LRF 3142 is unlike that seen in any of the aforementioned crocodyliform groups. Therefore, the possibility of crocodyliform affinities for LRF 759 and LRF 3142 is excluded here in favour of a group of terrestrial vertebrates whose teeth more closely match their distinct characteristics (see below).

Inclusion within Pterosauria

Australian pterosaur teeth are known only from in situ dentary and maxillary teeth of Mythunga camara (Molnar & Thulborn, 2007, fig. 2) and an isolated tooth associated with the rostral portion of an ornithocheiroid mandible (Fletcher & Salisbury, 2010, figs. 3I–3J), both from the Lower Cretaceous of central Queensland. All teeth have elongated conical crowns with heights averaging approximately 20 mm and an oval basal cross-section. The teeth of Mythunga camara are slightly distally recurved and bear an enamel ornamentation of irregularly-spaced longitudinal grooves on the basal two thirds of the crown. The single tooth described by Fletcher & Salisbury (2010) is devoid of any enamel ornamentation.

Pterosaur teeth are infrequently preserved with cranial material and readily dislodge from the alveoli post mortem. Isolated teeth are more common, but comprise a relatively small proportion of the terrestrial vertebrate fossil record during the Mesozoic. The overwhelming majority of pre-Cretaceous pterosaurs had toothed jaws, but during the Cretaceous a number of pterosaur lineages independently lost dentition either partially or completely. Among these clades are the nyctosaurids, pteranodontids, chaoyangopterids, tapejarids and azhdarchids, and as such they cannot be considered as candidates for the Lightning Ridge teeth.

Ctenochasmatidae is the only clade of archaeopterodactyloids to have survived into the Cretaceous. The dentition of ctenochasmatids consists of a large number of recurved, elongated, needle-like teeth in both the upper and lower jaws (e.g., Huanhepterus, Gegepterus, Moganopterus; Dong, 1982; Wang et al., 2007; Lü et al., 2012). This dental morphology is taken to an extreme by Pterodaustro in which approximately 1,000 bristle-like teeth lined the jaws (Chiappe & Chinsamy, 1996). Among ornithocheiroids, istiodactylids have ‘lancet-shaped’, or triangular, labiolingually-compressed crowns (Witton, 2012). Carinae may either be present mesially and/or distally (e.g., Nurhachius, Istiodactylus sinensis; Wang et al., 2005; Andres & Ji, 2006) or absent entirely (e.g., Hongshanopterus, Wang et al., 2008). Dsungaripterids are the only azhdarchoid pterosaurs that are not edentulous. Dsungaripterid dentition consists of apicobasally-short crowns with obtusely-pointed apices, restricted to the posterior part of the upper and lower jaws (Young, 1964; Unwin, 2003).

The teeth of anhanguerians (Rodrigues & Kellner, 2013) are typically slightly labiolingually-compressed with an elliptical basal cross section. The posterior dentition in some taxa is characterised by low, labiolingually triangular crowns (e.g., Cearadactylus atrox, Guidraco venator; Wang et al., 2012; Vila Nova et al., 2014). The crowns are slender and elongate, though not to the extent seen in ctenochasmatids. A slight distal recurvature of the crown is common to most anhanguerians (e.g., Anhanguera araripensis, A. piscator, Siroccopteryx, Ludodactylus; Wellnhofer, 1985; Mader & Kellner, 1999; Kellner & Tomida, 2000; Frey, Martill & Buchy, 2003) although in some taxa the crowns are recurved only apically or not at all (e.g., Cearadactylus atrox; Vila Nova et al., 2014). A slight lingual curvature is also present in some anhanguerian teeth (e.g., Wellnhofer & Buffetaut, 1999; Averianov, 2007) but can become very strong as in the posterior dentition of A. araripensis in which the apices can point directly lingually (Wellnhofer, 1985, fig. 7). Both mesial and distal carinae are present in some taxa (e.g., A. santanae; Wellnhofer, 1985) but are absent in others (e.g., A. robustus, Siroccopteryx; Wellnhofer, 1987; Wellnhofer & Buffetaut, 1999). The enamel on the crowns is typically ornamented by longitudinal grooves (e.g., A. robustus, A. piscator, Mythunga, Guidraco; Wellnhofer, 1987; Kellner & Tomida, 2000; Molnar & Thulborn, 2007; Wang et al., 2012). The teeth of A. araripensis appear to lack any longitudinal grooves (Wellnhofer, 1985); however, only the posterior dentition of this taxon is currently known and it is possible that its anterior teeth were similar to those of its congenerics.

LRF 759 and LRF 3142 bear little resemblance to the needle-like dentition of ctenochasmatids, the ‘lancet-like’ dentition of istiodactylids, or the blunt triangular dentition of dsungaripterids. However, a comparison of anhanguerian dentition to the Lightning Ridge teeth demonstrates a compelling similarity. The teeth of the probable anhanguerian Mythunga camara are similar in size and shape to the Lightning Ridge teeth, and in the case of LRF 759 have a similar enamel ornamentation of discontinuous longitudinal grooves (Molnar & Thulborn, 2007). The longitudinal grooves on the crown of LRF 759 are not as deeply impressed as in some species of Anhanguera (e.g., A. robustus, A. piscator; Wellnhofer, 1987; Kellner & Tomida, 2000) and are more similar to those described from other anhanguerians (e.g., Guidraco; Wang et al., 2012). The small degree of lingual curvature of the crowns is also observed in the anterior dentition of some species of Anhanguera (e.g., A. araripensis, A. santanae; Wellnhofer, 1985).

Isolated anhanguerian-like teeth described as Morphotype III from Morocco (Wellnhofer & Buffetaut, 1999, fig. 8), Morphotype 3 from Spain (Sánchez-Hernández, Benton & Naish, 2007, fig. 5) and ZIN PH no. 41/43 of Averianov (2007, figs. 1d–1f) share with LRF 3142 elongate and slightly labiolingually-compressed crowns with an oval basal cross-section, very slight distal recurvature, and unserrated carinae on both mesial and distal edges. Pterosaur teeth recovered from the middle Cretaceous Alcântara Formation and the Lower Cretaceous Recôncavo Basin, both in Brazil (Elias, Bertini & Medeiros, 2007; Rodrigues & Kellner, 2010), are morphologically similar in comparison to LRF 3142 with the exception of the absence of carinae. The distribution and phylogenetic significance of carinae in pterosaur dentition has yet to be examined in detail (Rodrigues & Kellner, 2010). It is uncertain whether the absence of characteristic pterosaurian surface ornamentation in LRF 3142 is genuine or a result of taphonomic processes. However, isolated pterosaur teeth without observable longitudinal grooves have been documented (e.g., Rodrigues & Kellner, 2010, fig. 3). Previously reported isolated anhanguerian-like teeth appear to be much longer than either LRF 759 or LRF 3142 (e.g., Wellnhofer & Buffetaut, 1999; Averianov, 2007). The morphology of both teeth appears to indicate that they were positioned towards the anterior end of their respective tooth rows, where the crowns are longer and more elongate in comparison to the more posterior crowns. However, the apical ends of both teeth are eroded, and the gentle degree of curvature of the mesial and distal margins in lateral perspective indicates that the apices would have been much longer prior to their taphonomic loss. Furthermore, the teeth are of a similar length to those preserved in the jaws of Mythunga camara (Molnar & Thulborn, 2007, figs. 2, 4).

Despite recent revisions of taxa and specimens that have historically been referred to Ornithocheirus and closely-related taxa (Unwin, 2001; Rodrigues & Kellner, 2008; Rodrigues & Kellner, 2013), many ‘ornithocheirids’ are still known from only partial and fragmentary remains and which lack diagnostic cranial material, including teeth. The continuing uncertainly surrounding the affinities of these remains hinders a comprehensive assessment of anhanguerian phylogeny, and in lieu of such assessments the isolated teeth cannot presently be referred to a clade less inclusive than Anhangueria.

Significance for Australian pterosaur diversity

This account represents the first description of pterosaur material from New South Wales, and permits the recognition of a new occurrence of this group of otherwise rare and poorly-known reptiles in Australia. Two pteranodontoid pterosaur taxa are currently recognised in Australia: Mythunga camara (QM F18896); and Aussiedraco molnari (QM F10613), both from the Lower Cretaceous of central Queensland. The jaw fragment WAM 68.5.11 (Kear, Deacon & Siverson, 2010) and partial mandible QM F44423 (Fletcher & Salisbury, 2010) possibly represent distinct ornithocheiroid taxa—the former based on its temporal separation from the aforementioned Queensland taxa and the latter from the distinct morphology of the mandibular symphysis (Kellner, Rodrigues & Costa, 2011). In addition, the azhdarchid ulna (WAM 60.57) from the Late Maastrichtian of Western Australia (Bennett & Long, 1991), and the ctenochasmatoid humeral fragment (QM F42739) (Fletcher & Salisbury, 2010) most likely represent distinct Australian pterosaur taxa. The remainder of the partial pterosaur material from Queensland and Western Australia may pertain to one or more of the aforementioned Early Cretaceous pterosaur taxa, or may represent new taxa that cannot be confidently identified. Thus it would seem reasonable to assume that at least six pterosaur taxa were present in Australia during the Cretaceous.

Although dentition may be diagnostic for particular pterosaur clades (see above), in the absence of articulated or associated skeletal material they typically are insufficient for identification of the tooth-bearer to a specific or generic level. In Australia, this problem is exacerbated by the scarcity of pterosaur remains to which the teeth described here can be compared. It is currently not possible to determine with certainty whether the Lightning Ridge teeth belong to one of the named or unnamed but potential Australian pterosaur taxa, or whether they constituted the dentition of a taxon that is yet to be discovered. Furthermore, given the subtle observed morphological differences between the two teeth (e.g., degree of recurvature, presence of carinae, enamel ornamentation, etc.) it is also uncertain whether the two teeth are derived from a single taxon or separate taxa. Further finds are needed in order to evaluate whether these differences are indicative of the presence of more than one pterosaur taxon, or whether taphonomic or other processes have affected the appearance of the tooth crowns.

The identification of anhanguerian teeth from the Griman Creek Formation is consistent with the reports of anhanguerid-like and ‘ornithocheirid’ skeletal material from the Early Cretaceous of Queensland (Molnar & Thulborn, 1980; Molnar & Thulborn, 2007; Fletcher & Salisbury, 2010) and the cosmopolitan distribution of ornithocheiroids at this time (Upchurch et al., 2015). The similarities in morphology are further supported by the similarities in palaeoenvironments. The Queensland pterosaur material from the Toolebuc and Mackunda formations of the Eromanga Basin were deposited in shallow waters near the central part of the Eromanga Sea during the early to middle Albian (Fletcher & Salisbury, 2010). Similar conditions prevailed during the middle Albian in the vicinity of present day Lightning Ridge. The occurrence of anhanguerid-like pterosaurs in near-shore and shallow water environments in Australia appears to correlate with the presumed diet of fish and other aquatic organisms that has been inferred for some anhanguerids (Kellner & Tomida, 2000), and which is evident from the slender, elongate and apically acute tooth crowns.

Conclusion

Isolated teeth excavated from the Lower Cretaceous Griman Creek Formation at Lightning Ridge, New South Wales, are identified as pertaining to pterosaurs. The oval basal cross-section, slight distal recurvature, irregularly-striated enamel ornamentation, and slender crowns bear a striking similarity to those of anhanguerian pterosaurs. This represents the first description of pterosaurs from New South Wales and contributes to the growing diversity of vertebrates from the Griman Creek Formation. The isolated remains cannot be conclusively assigned to any known pterosaur taxon, although their presence is consistent with the known record of anhanguerid-like pterosaurs from the contemporaneous Toolebuc Formation of central Queensland. The simultaneous presence in New South Wales and Queensland of anhanguerian pterosaur remains in sediments displaying characteristics of shallow-water lagoonal and lacustrine depositional environments indicates likely similarities in life habits of these pterosaurs. Further finds and descriptions of Australian pterosaurs are necessary to further characterise the diversity of this poorly understood group of reptiles both locally and in Australia as a whole.

Our thanks to Graeme and Christine Thomson for donating LRF 759, and Tony Morris and Gary Roberts for donating LRF 3142, to the Australian Opal Centre through the Australian Government’s Cultural Gifts Program. Jenni Brammall, Manager of the Australian Opal Centre, is also thanked for allowing access to the specimens and providing resources to facilitate their study while in Lightning Ridge. Detailed reviews by Alexander Kellner and Taissa Rodrigues improved the quality of the manuscript and we thank them for their efforts.

Institutional abbreviations

LRF (Australian Opal Centre, Lightning Ridge)

QM (Queensland Museum, Brisbane)

WAM (Western Australian Museum, Perth)

ZIN (Zoological Institute of the Russian Academy of Sciences, St. Petersburg)

Additional Information and Declarations

Competing Interests

Author Contributions

Data Availability

The authors declare there are no competing interests.

Tom Brougham conceived and designed the experiments, analyzed the data, wrote the paper, prepared figures and/or tables.

Elizabeth T. Smith reviewed drafts of the paper.

Phil R. Bell conceived and designed the experiments, reviewed drafts of the paper.

The following information was supplied regarding data availability:

The raw data is included in Table 1.

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
