# Peer review of "Isolated teeth of Anhangueria (Pterosauria: Pterodactyloidea) from the Lower Cretaceous of Lightning Ridge, New South Wales, Australia"

_PeerJ, doi:10.7717/peerj.3256_

## Round 0.1 · original submission · Minor Revisions

Overall, only minor changes (outlined in the reviewers' comments) are required. I have nothing to add at this time beyond what the reviewers note.

·

Basic reporting

The ms is well-written and references are up-to-date. I only would point out that there are different hypothesis out there when it comes of pterosaur relationships - not only the one listed by them. This is a minor point and does not invalidate their research.

Experimental design

Nothing to add here.

Validity of the findings

A new occurrence for pterosaurs that justifies the publication of this ms.

Additional comments

Pterosaurs have been found in several deposits around the world, but in some areas they remains quite scarce. This is definitively the case of Australia.
The present ms describes two teeth attributed to the clade Anhangueria. As presented in the ms., this is the first record of those flying reptiles published from New South Wales and this only is worth for pterosaur researchers.

Anhangueria might be a much larger group than presently regarded and the occurrence of this material might indicate this, particularly when more pterosaur material from this region comes to light.

The only problem I see is that the teeth seems to be a little bit short when compared to anhanguerid-like pterosaurs. Perhaps this might be so due to the fact that they are incomplete. Otherwise, I do not know of any other reptile group to which this material could be assigned. The authors might like to make this a little bit more clear in their ms.

The authors are also entitled to follow any phylogenetic hypothesis that is out there, but perhaps might like to acknowledge that there are competing ideas about pterosaur phylogeny - not only Andres et al.

·

Basic reporting

This is a well-written manuscript. Language is appropriate and clear. The introduction provides a background to the context of the manuscript, and there is extensive citation to the relevant literature.

One funding agency is thanked in the Acknowledgements section; this citation should be made elsewhere.

Figures are relevant and of high quality. However, even though figure 2 is very interesting as it provides an overview of Australian pterosaur findings x time x geological context, it is not cited in the text. Please do include a reference to this figure.

Legends of figures 3 and 4 acknowledge that the identification of lingual and labial views is tentative, by using “labial(?)” and “lingual(?)”. The same should be used in the legend of figure 5. Other than that, all figures are well labeled and described.

No raw data was provided and none is needed.

Experimental design

The research is original and within the scope of PeerJ. It regards the identification of isolated fossil teeth from Australia and, following a broad comparison to several vertebrate groups (fishes, plesiosaurs, ichthyosaurs, dinosaurs, crocodiliforms), the authors conclude that they belong to anhanguerian pterosaurs. Methods are described in detail and the study is replicable.

Validity of the findings

The data is robust and sound. Conclusions are well stated and supported by the results.

Additional comments

1. This is a straight forward, well written work that I believe will be of interest for a broader audience, i.e., any paleontologist dealing with isolated vertebrate remains found during fieldwork. With that in mind, I suggest that the authors include photos or drawings of teleost, plesiosaur, ichthyosaur, theropod, and crocodyliform teeth that superficially resemble the ones described in this work.

2. In the description, the terms lingual and labial are often used. I could not understand, however, how was this identification made, as differences between these sides are unreported and as upper and lower teeth are identical in anhanguerians. In two of the legends to the figures, the terms are used followed by a question mark [“lingual(?)”, “labial(?)”]. It should be clearly stated in the text how this identification was made. In any case, the use (or not) of question marks should be the same both in the text and in the legends.

3. I congratulate the authors on using standardized terminology. It would be interesting to continue this trend and provide a short explanation on the differences between “striations”, “ridges”, and “wrinkles”.

4. Regarding the absence of striations in A. araripensis: the teeth described and figured by Wellnhofer are the caudal, smaller ones. To the best of my knowledge, all anhanguerian rostral teeth have striations. So it seems likely that the rostral teeth of A. araripensis also had striations but, as the holotype doesn’t have the rostrum preserved (and referred specimens don’t have the teeth preserved), we simply cannot see them. It would be a matter of incompleteness rather than an actual difference between species, as suggested in the text.

5. There is one work in a similar context - the identification of isolated anhanguerian teeth - that the authors might find useful: Rodrigues and Kellner (2010). Note on the pterosaur material described by Woodward from the Recôncavo Basin, Lower Cretaceous, Brazil. Revista Brasileira de Paleontologia 13(2):159-164. DOI: 10.4072/rbp.2010.2.08. Available at: http://www.sbpbrasil.org/revista/edicoes/13_2/Artigo%208%20-Rodrigues%20&%20Kellner.pdf

There is also a paper on isolated crocodyliform teeth from the same deposit as above: Souza et al. (2015) Taxonomic revision of Thoracosaurus bahiensis Marsh, 1869, a supposed Gavialoidea (Reptilia, Crocodylia) from Cretaceous deposits of the Recôncavo Basin, Brazil. Revista Brasileira de Paleontologia 18(3):565-568. doi: 10.4072/rbp.2015.3.17. Available at: http://www.sbpbrasil.org/assets/uploads/files/17_Souza_et_al_pg565a568_web.pdf. The difference in the ornamentation of both taxa is remarkable and the authors might find it useful.

6. Other small remarks are marked in the PDF.

---

## Round 0.2 · Minor Revisions

Thank you for your close attention to the reviewer comments. The manuscript is nearly entirely ready to go, but one final issue that should be addressed concerns the silhouette for Anhangueria in Figure 2. PeerJ is published under a CC-BY license (not CC-BY-SA, which is a different set of licensing parameters), so CC-BY-SA is not a suitable license for image components. Fortunately, I note that the silhouette is derived from a CC-BY paper (Claessens LPAM, O'Connor PM, Unwin DM (2009) Respiratory Evolution Facilitated the Origin of Pterosaur Flight and Aerial Gigantism. PLoS ONE 4(2): e4497. doi:10.1371/journal.pone.0004497). Thus, the easiest solution would be to redraw the silhouette from that source, and then cite the Claessens et al. paper. I realize this is a bit convoluted, but hopefully it would take very minimal time and would ensure that everything is above-board for image use.

---

## Round 0.3 · accepted · Accept

Thank you for your quick attention to the last edit request. Your paper is good to go from my standpoint.